# Numerical Simulation of Local Buckling of Submarine Pipelines under Combined Loading Conditions

**DOI:** 10.3390/ma15186387

**Published:** 2022-09-14

**Authors:** Wenxian Su, Jie Ren

**Affiliations:** School of Energy and Power Engineering, University of Shanghai for Science and Technology, Shanghai 200093, China

**Keywords:** submarine pipelines, global buckling, initial imperfection, local buckling

## Abstract

Submarine pipelines are prone to developing flaws, such as ellipticity and depression during the manufacture, burying, and use processes. The local buckling characteristics of submarine pipelines with initial imperfection must be studied since the initial imperfection have an impact on local pipeline buckling. In this study, the local buckling of submarine pipelines with varying depression depths and ellipticity is simulated using the finite element program ABAQUS, and defect sensitivity of submarine pipelines with varying shape ellipticity, varying depression depths, and varying pipe radius-thickness ratios is examined. Meanwhile, research is being conducted on the combined load buckling of a submarine pipeline with initial imperfection caused by bending, axial force, and external hydrostatic pressure. The results indicated that the critical external pressure of the pipeline is sensitive to the imperfection, although the buckling propagation pressure is not. The buckling morphology is influenced by the shape and size of the imperfection. Additionally, the ability to withstand external hydrostatic pressure of the pipeline reduces after it has been bent.

## 1. Introduction

The primary component of the offshore oil and gas field development and production system, the submarine pipeline is crucial to the transportation of offshore oil and gas. The advantages of a submarine pipeline are continuous oil delivery, high oil delivery rates, and low environmental impact. Additionally, the submarine pipeline laying process is fast, production is efficient, management is simple, and the cost of operation is affordable. The local buckling of submarine pipelines is very easy to happen under the influence of intense water pressure because of initial imperfection. The elliptical cross-section, local dog-bone shape, or fold in the tube wall are characteristics of the buckling morphology. As soon as this buckling instability forms, it will quickly spread throughout the length of the pipeline, leading to large-area buckling that will impair the fluidity of the entire submarine pipeline system [1]. In the process of designing and manufacturing submarine pipelines, the local buckling problem has grown to be a significant issue that requires research.

The geometric size and material property distribution are uneven to some extent throughout the production and manufacturing process of submarine pipeline, which causes the pipeline to have initial imperfection. If corrosion or collision occurs during the laying and usage of the pipeline, the imperfection will be further worsened, and this will significantly limit the pipeline’s ability to resist buckling collapse [2].

Domestically and internationally, researchers have taken a series of studies on the local buckling of submarine pipelines since the buckling propagation phenomena was discovered by Buttelle Columbus Laboratory [3] in 1970. Numerical simulation was utilized by Ramasamy et al. [4] to examine the buckling propagation for pipes of various geometric sizes, confirming the accuracy of finite element software simulation. Pournara et al. [5] investigated the pipeline failure modes under cyclic bending load and cyclic pressure load by using test and numerical modeling and presented a method for determining the residual strength of dented pipeline. Run Liu [6] and others proposed an elastic-plastic three-dimensional explicit numerical method to study the global transverse buckling of single defect and double defect pipes. Wenbin Liu [7] and others used the numerical simulation method to analyze the global buckling process of the pipeline with two initial imperfections. Degenhardt et al. [8,9] developed methods to accurately compare the overall impact of the axial buckling load to various imperfection types. The results indicate that imperfections in the overall thickness play the largest role in determining the overall axial buckling load of imperfection sensitive unstiffened cylinders, followed closely by loading imperfections. General material flaws have little effect on the axial buckling load. Thickness imperfections must be taken into account during the design process because they have a greater impact on the axial buckling load than other flaws. The findings indicate that a pipeline with two initial imperfections may experience superposition buckling. The buckling collapse of pipes with a small radius-thickness ratio were explored by Zhang Rixi [10]. He proposed the correction formula for the limit load of pipes with a small radius-thickness ratio based on the classical theory and the numerical solution. The pipeline model was created using ABAQUS and calculated using the Riks method by Li Xinzhong et al. [11]. They also examined the buckling collapse of the submarine pipeline and fitted the ultimate load-carrying formula. By using the nonlinear finite element method, Tian Long [12] investigated the collapse mechanical properties of pipelines with initial imperfections and the buckling properties of submarine pipelines under external pressure. Yu Jianxing [13] et al. explored the buckling failure of submarine pipelines with initial ovality imperfection under the combined action of bending moment and water hydraulic. To examine external pressure and force couple combination load, Chen Yanfei et al. [14] used the numerical simulation approach. The extreme bending moment of the depression submarine pipeline and the local buckling failure mechanism were employed to conduct sensitive analysis of the parameters impacting the limit bending moment of the pipeline.

The local buckling of depressed submarine pipelines under the combined action of complex loads is relatively rare, despite the fact that researchers from both domestic and foreign universities have conducted a number of studies on the subject. Most of these studies have focused on intact pipelines with single defect and single load or double load forms. Based on the aforementioned factors, this study investigates the local buckling failure of submarine pipelines and the buckling and collapse of defective submarine pipelines under complex loads under the combined action of bending, axial force, and external hydrostatic pressure. The findings of the study can serve as a theoretical foundation for work that is conducted in practice.

## 2. Critical Load of Local Buckling of Pipeline

### 2.1. Pipeline Buckling under External Pressure

In contrast to global buckling, local buckling causes the pipe to deform in a narrow range. Moreover, local buckling is a significant factor leading to buckling propagation. Pipe geometry is invariably inaccurate during the manufacturing, transportation, and installation processes. The most common is the ovalization of the pipe section, which greatly weakens the ability of the pipe to maintain its own shape and position balance. The initial degree of ellipticity is introduced to describe the ovality:(1)$${∆}_{0}=\frac{D_{max}-D_{min}}{D_{max}+D_{min}}$$

Among them: $D_{max}$ is the maximum possible diameter of the pipeline; $D_{min}$ is the minimum possible diameter. Another common initial imperfection is thickness nonuniformity, which is generally quantified by the thickness nonuniformity coefficient: (2)$$\Xi_{0}=\frac{t_{max}-t_{min}}{t_{max}+t_{min}}$$
where: $t_{max}$ is the maximum possible thickness; $t_{min}$ is the minimum possible thickness.

A pipeline’s ovality is a type of pipeline defect that can reduce a pipeline’s buckling external pressure. When the external pressure reaches a certain value, the ring bending stress and membrane stress produce plastic hinges at the four ends of the long axis and the short axis, causing pipeline instability and collapses. For the thin-walled steel tube with initial ellipticity, Timoshenko obtains the equation through theoretical derivation:(3)$$P_{cr}^{2}-\left[P_{p}+\psi P_{e}\right]P_{cr}+P_{p}P_{e}=0$$

Buckling pressure can be obtained by solving this equation:(4)$$P_{cr}=\frac{1}{2}\left\{\left(P_{p}+\psi P_{e}\right)-{\left[{\left(P_{p}+\psi P_{e}\right)}^{2}-4P_{p}P_{e}\right]}^{1/2}\right\}$$
where:(5)$$\psi =1+3{∆}_{0}\frac{D}{t}$$

The plastic strain of the material is disregarded in Formula (3). In actuality, submarine pipelines typically have walls that range in thickness from 10 to 35, making them vessels with thick walls. When determining the buckling pressure, the impact of material nonlinearity cannot be disregarded.

### 2.2. Pipeline Buckling under Combined Action of Bending Moment and External Pressure

For the buckling of pipes under pure bending moment, Brazier gives the extreme bending formula:(6)$$M_{m}^{e}=\frac{M^{o}EJt^{o}}{R}$$
where:$$J=\pi R^{3}t;\;t^{o}=\frac{1}{R}\sqrt{\frac{D_{1}}{Et}};\;D_{1}=\frac{Et^{3}}{\left[12\left(1-\mu^{2}\right)\right]}\;M^{o}=1.089;\;R=\frac{D}{2}$$

Here, D is the nominal diameter and t is the nominal thickness.

Later, researchers corrected $M^{o}$ after a lot of research, and most of the values they gave were between 2.4% and 5.9% smaller than those given by Brazier.

Timeonshenko and Gere put forward the classical theoretical calculation formula of elastic instability pressure of the infinite cylindrical shell [15]: (7)$$P_{C}=\frac{2E}{1-\mu^{2}}{\left(\frac{t}{D}\right)}^{3}$$

When the pipe is subjected to both bending force and external pressure, the extreme value of the critical load is determined by formula [16]:(8)$$(\frac{M}{M_{m}^{\varepsilon }}{)}^{2}+\frac{P}{P_{c}}=1$$
where: $M_{m}^{\varepsilon }$ is the extreme bending moment of the pipeline under the pure bending moment calculated according to Formula (6); it is the elastic buckling pressure of the pipe under the action of external pressure calculated according to Formula (7). P and M are uniformly distributed external pressure and external bending moment, respectively.

Later, researchers used this foundation to introduce the initial imperfection factor and obtain a more applicable formula:(9)$${\left(\frac{M}{M_{m}^{e}}\right)}^{2}+\frac{1+3\lambda }{4}\left(\frac{P}{P_{c}}\right)=1$$
(10)$${\left(\frac{\kappa }{\kappa_{m}^{e}}\right)}^{2}+\frac{1+3\lambda }{4}\left(\frac{P}{P_{c}}\right)=1$$
where: κ is the bending curvature; λ is the initial imperfection factor.
(11)$$\lambda =1+2\frac{\omega_{0}}{\omega }$$

Here, ω represents the buckling waveform and $\omega_{0}$ represents the defect waveform

If the initial ovality coefficient is introduced, Formulas (9) and (10) can be approximately written as: (12)$$1.180\eta_{m}^{\lambda }{\left(\frac{M}{M_{m}^{e}}\right)}^{2}+\frac{1+3\lambda }{4}\left(\frac{P}{P_{c}}\right)=1$$
(13)$$1.288\eta_{\kappa }^{\lambda }{\left(\frac{\kappa }{\kappa_{m}^{e}}\right)}^{2}+\frac{1+3\lambda }{4}\left(\frac{P}{P_{c}}\right)=1$$
(14)$$\;\;\;\;\;\left\{\begin{array}{c} \eta_{m}^{\lambda }=\frac{0.1256}{\chi }{\left[\frac{1}{1-\chi -\left(\frac{3\lambda }{2-\frac{5}{6}}\right)\chi^{2}}\right]}^{2}\\ \eta_{\kappa }^{m}=\frac{0.2589}{\chi }\\ \chi =\frac{\sqrt{30\lambda -7.66}-3}{15\lambda -8.333} \end{array}\right.$$

## 3. Finite Element Analysis

### 3.1. Finite Element Model

Cross-section ovality and local depression are two of the introduced flaws.

The geometric model of a pipeline has been constructed. The finite element model of the submarine pipeline is shown in Figure 1; its overall length is 1000 mm. The ovality (Ratio of the difference between the long and short axes to the sum of the long and short axes) is set at 0.1%, 1%, and 1.5%. The length of the transition section from the elliptical section to the circular section is 50 mm. According to the symmetry, the total length of the transition section is 100 mm. The average length of the element for establishing the finite element model is 4 mm. When establishing the dented pipeline model, the method adopted is to cut the target surface with a cylindrical surface with a large radius to form a circular arc depression. The ratio of maximum depth to diameter is:(15)$$\;\Omega =\frac{\Delta D}{D}\times 100\%$$

The critical pressure is 2.2 MPa, and the pitch diameter of the pipeline is 41 mm, according to the theoretical Formula (7). The higher pressure value is required for the nonlinear buckling analysis. The incremental step is set at 0.05, and the external pressure is gradually loaded.

### 3.2. Pipe Material Parameters

The pipe portion within a specific length is chosen for simulation based on the analysis of pipeline up–floating buckling. Table 1 display the configured pipe parameters.

$D_{0}$ is the outer diameter of the pipe, t is the thickness of the pipe wall. This chapter assumes that the pipeline material is bilinear constitutive relation, E is Young’s modulus, $E_{t}$ is tangent modulus. The yield strength $\sigma_{y}$ of the pipeline is 90 MPa.

### 3.3. Unit Selection

The radius-thickness ratio of the pipeline is greater than 10, and the radial force is small for the other two directions, thus, it can be ignored. In addition, the buckling problem studied in this paper considers the influence of initial geometric imperfection and bending, without considering the influence of transverse shear deformation on the results. Based on the above factors, this paper chooses the S4R general-purpose three-dimensional shell element to calculate. S4R is a finite strain unit that allows the existence of shear strain. It should be noted that the normal direction of the shell element is exactly the opposite to that of the solid element. The normal direction of the three-dimensional shell element can be judged according to the right-handed rule.

### 3.4. Boundary Conditions

Because the pipeline’s modeling length is 1000 mm, tens of times the diameter, and the constraint conditions at either end have little impact on the middle portion. One end restricts the circumferential displacement while the other restricts the axial and circumferential displacement in order to facilitate.

## 4. Result Analysis and Discussion

### 4.1. Sensitivity Analysis of Local Buckling Defects of Submarine Pipelines

#### 4.1.1. Comparison of Simulation Results with Measured Values

In this section, the pipe with the radius-thickness ratio of 42 is selected for simulation.

Figure 2, Figure 3 and Figure 4 represent the deformation of the three different ellipticities. Due to the imperfect local structure, the first region that was affected by defects first deformed and developed on both sides. Figure 5 displays the various displacement load curves for pipelines with ovalities of 0.001, 0.01, and 0.15. When the ovality is 0.001, the critical external pressure is 2.26 MPa, which is slightly higher than the theoretical value; when the ovality is 0.01, it drops to 2.08 MPa, which is 5.5 percent lower than the elastic collapse pressure; when the ovality is 0.015, the critical external pressure is reduced to 2.01 MPa, with an 8.6 percent reduction rate. Figure 5 shows that local faults can cause the pipe to buckle. For pipes of various ovalities, the same pre-buckling equilibrium path and post-buckling equilibrium path can be observed.

Due to the initial imperfections in the pipeline, the critical pressure PI (also known as the initial pressure) is generally less than PC in Formula (7). Palmer and Martin [17] give the buckling propagation pressure of the pipe:(16)$$\;P_{PM}=\pi \sigma_{y}{\left(\frac{t}{D}\right)}^{2}$$

According to the formula, the buckling propagation pressure is far less than the critical pressure. In Figure 5, the pressure value of the gentle section on the right of the curve corresponds to the buckling propagation pressure. The combined value of the three pipes is approximately from 0.24 MPa to 0.25 MPa, which is significantly less than the critical pressure. Therefore, the pipe will propagate to both ends once local buckling occurs. The buckling propagation pressure is the control factor of the submarine pipeline design, which increases material and cost. The initial pressure represents the sudden springing stability of the pipeline and is extremely sensitive to defects, while the propagation pressure $P_{P}$ is not sensitive to defects.

#### 4.1.2. Effect of Depression on Critical Pressure

This section studies the effects of different pipe diameters, depression depths, and depression lengths. The pressure response of the pipe is shown in Figure 6, Figure 7, Figure 8 and Figure 9 when the pipe diameter is 41 mm, the depression length is 20 mm, and the depression rate is 1%, 1.5%, 2%, and 3%, respectively. The pressure response of pipe with a depression length of 20 mm is shown in Figure 10.

When the pipe diameter is 41 mm, the defect length is 50 mm, and the depression rate is 1%, 1.5%, 2%, and 3%, respectively, the pressure response of the pipe is shown in Figure 11.

Similar to the earlier situation, where the length is 20 mm, the depression depth mostly affects the initial pressure of pipe but has no effect on the post-buckling equilibrium path. When the depression length is 20 mm, different depression depths correspond to various post-buckling equilibrium paths, and the pipe deformation is also various, as shown in Figure 12. The buckling equilibrium paths of pipelines with different depression lengths are arranged in a diagram to make comparisons easier, as shown in Figure 5. The critical pressure decreases as depression length lengthens, but the post-buckling equilibrium pressure rises to varying degrees. When the depression length is minor, the pipeline buckles into the shape of a dog bone with the upper and lower (or) left and right sides simultaneously close to the middle. When the depression length is sufficient, the defected side experiences the first deformation. As a result of external pressure, the depression area lengthens and deepens.

#### 4.1.3. Sensitivity of Different Radius-Thickness Ratio to Defects

In this section, the pipe with a radius-thickness ratio of 27 is utilized for calculation, and the calculation results are compared to the pipe with a radius-thickness ratio of 41. The critical pressure for a pipeline with a radius-thickness ratio of 27 mm is 7.7 MPa, according to theoretical calculations.

As can be observed from Figure 13, the critical pressure is almost 19% less than the theoretical value at 6.24 MPa when the depression ratio is 0.01, and it is 5.45 MPa when the depression ratio is 0.015. For the pipe with radius-thickness ratio of 41, these two values are close to the theoretical value. This indicates that the smaller the radius-thickness ratio, the more sensitive the pipeline is to local defects. According to the defect sensitivity analysis, the initial pressure (critical pressure) of the pipeline is sensitive to the initial imperfections. The local buckling of the pipe is an important factor causing the propagation of destructive buckling. It is necessary to analyze any factors that may cause local buckling.

### 4.2. Nonlinear Local Buckling of Submarine Pipelines under Combined Axial Force, Bending Moment, and External Pressure

#### 4.2.1. Analysis Model

Here, d stands for the formation of a localized initial wrinkle defect on the compression side of the pipeline’s center position. The fold shape control equation is [18]:(17)$$\bar{\omega }=-\frac{D}{2}\left[a_{0}+a_{i}\cos(\frac{\pi x}{N\lambda })\right]\cos\left(\frac{\pi x}{\lambda }\right)0\leq x\leq N\lambda$$
wherein, the half wavelength of defect is λ =0.165D, the half wave number is N=11, and the amplitude is $a_{0}=0.0025$, $a_{i}=0.2a_{0}$. 

The wrinkle defect model is shown in Figure 14.

A quarter model is established to facilitate the addition of boundary conditions. According to Figure 15, the two straight edge segments L1 and L2 limit displacement in the X direction and rotation in the Z direction; L4 is the pipe’s symmetry plane, which permits movement in the Y direction while restricting displacement in the Z direction and rotation in the X direction; L3 is the end, where an axial force and bending moment are applied to limit displacement in the Y direction. Figure 16 depicts the pipeline’s finite element model and boundary conditions.

Additionally, the defect that affects the stability of external pressure must be set up in order to compute the buckling of pipes under external pressure. As shown in Figure 17, the buckling mode of the pipe under external pressure in the middle part is produced using the eigenvalue approach in this research, and the first mode is extracted. The defect factor set in the nonlinear analysis is 0.02.

Step-by-stage loading is necessary to analyze how global bending affects local buckling: in the first step, the axial force is gradually loaded at the far end and released as the pipe floats up for buckling. The axial force can be thought of as being minimal. Additionally, axial force has less of an impact on local buckling than the bending moment, and the axial force applied in this paper is unified as 0.4 Py; the bending moment is gradually increased in the second step, provided that the axial force in the first step remains unchanged; step 3 gradually loads the external pressure based on the results of the first two phases. The general nonlinear algorithm is used in the first and second analysis phases of the computation, and the third analysis step uses the risk arc length approach. This technique allows for the modification of the second step bending moment and the determination of various critical external pressures. In this work, the nonlinear buckling of the pipeline is examined for bending loads of M = 0 Mp, M = 0.2 Mp, M = 0.4 Mp, and M = 0.8 Mp.

#### 4.2.2. Pressure-Displacement Curves of Pipelines Subjected to Different Bending Moments

The study of the influence of bending is actually a study of the critical pressure of the pipe under various bending moments because the bending curvature of the pipe has a linear relationship with the applied bending moment in the elastic range. The correlation between pressure and displacement is depicted in Figure 18. The pipeline’s buckling equilibrium path is similar to that when just external pressure is applied when M = 0. The initial displacement is nil, the post-buckling equilibrium path is mild, and the buckling happens when the pressure reaches the critical value. When M = 0.2 Mp, it can be seen from the figure that the displacement is almost zero when the pressure is 0, indicating that the bending moment does not cause large displacement of the reference point. However, during the pressure rise, the position of the reference point changes to a certain extent, corresponding to the rising section on the left side of the curve in the figure. If the bending moment is increased, when the second load step stops and the third load step does not start, the pipe will have a large node displacement. For instance, the node displacement is 5 mm when M = 0.8 Mp. The post-buckling equilibrium path is not smoothed-out when there is a bending moment, but it does have a slight downward slope. This is because after buckling, although the external pressure decreases gradually, the effect of bending moment still exists, and the pipe continues to deform. When calculating with the arc length method, it is inevitable to reduce the external pressure in order to maintain the balance of the pipe.

#### 4.2.3. Effect of Bending on Critical Pressure

Table 2 lists the critical buckling pressure and reduction rate for the pipe at various bending moments. It is evident that the critical pressure is less impacted by the bending moment when the bending moment is small. The critical pressure decreases by between 10.38 and 18.40 percent when the bending moment surpasses 0.4 Mp. This means that when the pipeline is global buckling, the critical external pressure lowers. In designing pipes, buckling interaction must be taken into account. The effect of axial force on the critical load is not immediately apparent; hence, it is certain that if the pipeline has excessive bending when the global buckling occurs, the critical pressure of local buckling will be affected. After the actual pipeline is bent, under the influence of local defects, the ability of the compression side to resist deformation is weakened, resulting in the decline of the stability of the pipeline.

## 5. Conclusions

(1) The finite element model of a submarine pipeline is developed using the finite element analysis program ABAQUS, and the susceptibility of local buckling to initial imperfection is investigated. The findings demonstrate that, for pipes of different ellipticity: the pre-buckling equilibrium path and the post-buckling equilibrium path are the same. They also demonstrate that the buckling of pipes is sensitive to local defects, the critical external pressure of pipes is sensitive to defects, and the buckling propagation pressure is insensitive to defects. Buckling shape is influenced by the magnitude and shape of the defect. For various depression lengths: The critical pressure decreases as the depression length increases, yet the pressure of the rear flexion balance increases to varying degrees. When the defect length is small, the pipeline buckles in such a way that the upper and lower (or) left and right sides are near the center simultaneously and the cross-section has the form of a dog bone. When the depression is long enough, it begins to distort on the damaged side first. The depression’s location lengthens and deepens in response to external pressure. For various radius-thickness ratios, the pipeline is more susceptible to local defects the smaller the radius-thickness ratio. The buckling type of the pipeline is impacted by the radius-thickness ratio.

(2) The local part’s impact of the submarine pipeline’s overall floating buckling is simplified as the local part’s impact of the bending moment. The influence of the bending moment on critical pressure is analyzed. When the pipeline is bent as a whole, it is discovered that the critical external pressure decreases. Pipeline design must take interaction of buckling into account, and axial force’s effect on critical load is not immediately apparent. As a result, excessive bending will reduce the bearing capacity of external hydrostatic pressure and will have an impact on the critical pressure of local buckling if the pipeline is vulnerable to global buckling.

## Figures and Tables

**Figure 1 materials-15-06387-f001:**
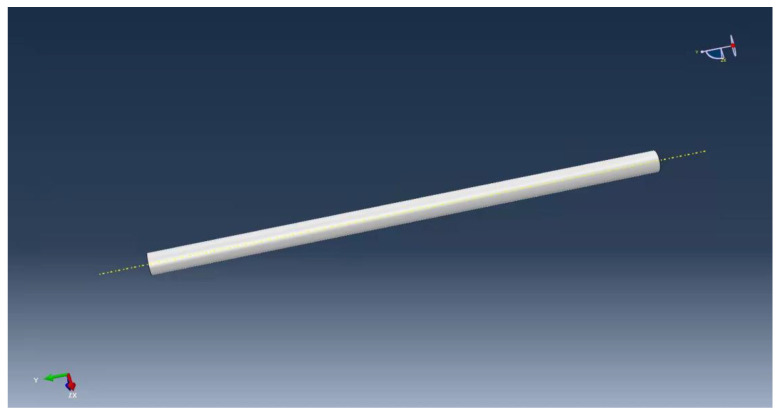
The finite element model of the submarine pipeline.

**Figure 2 materials-15-06387-f002:**
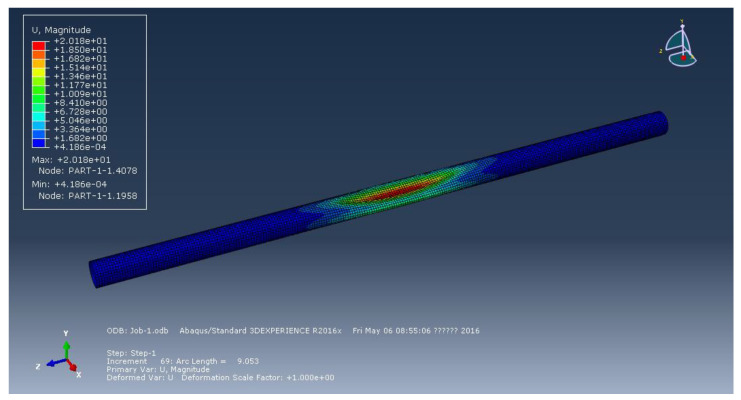
Deformation of pipeline at ellipticity of 0.001.

**Figure 3 materials-15-06387-f003:**
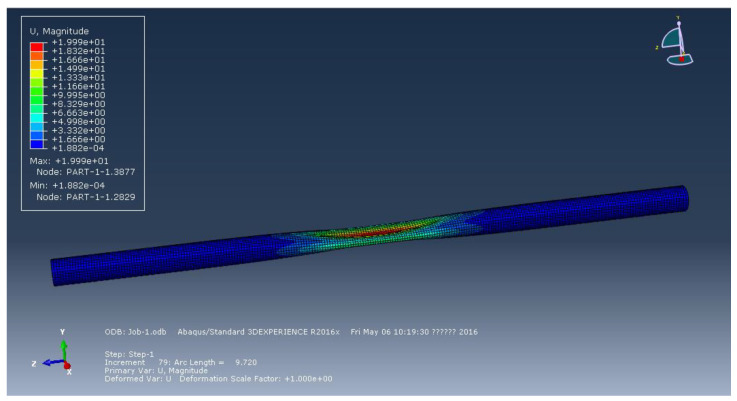
Deformation of pipeline at ellipticity of 0.01.

**Figure 4 materials-15-06387-f004:**
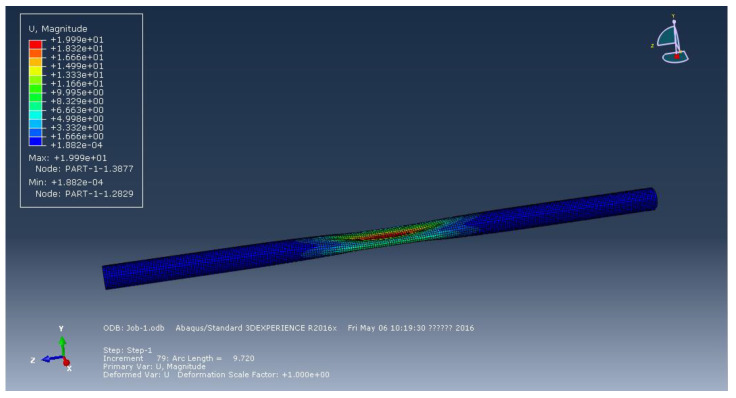
Deformation of pipeline at ellipticity of 0.15.

**Figure 5 materials-15-06387-f005:**
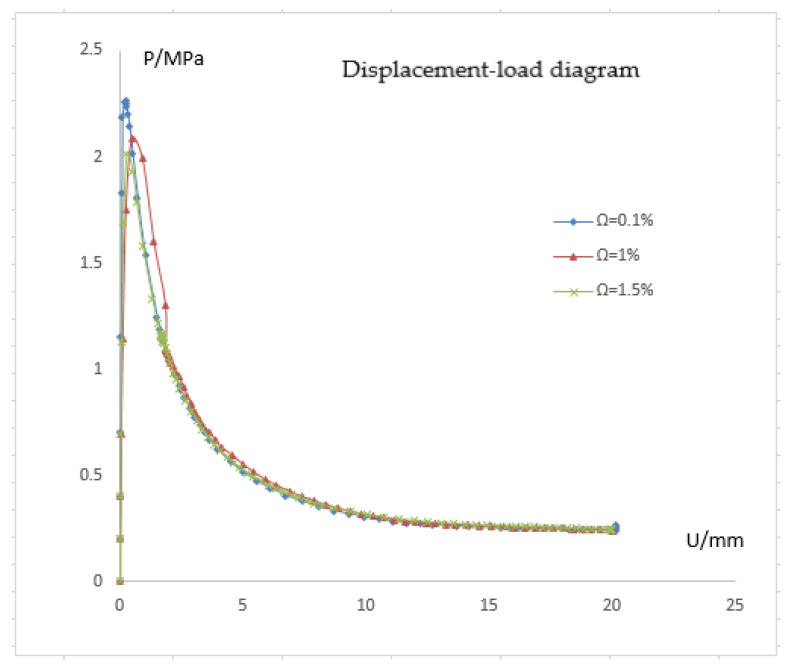
Pressure response curves of pipes with different ellipticity.

**Figure 6 materials-15-06387-f006:**
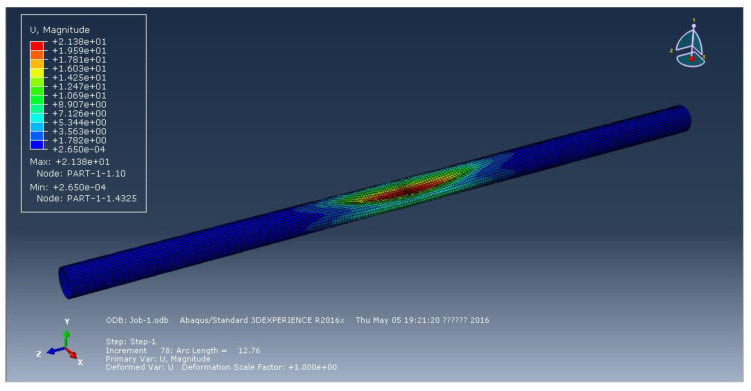
Distortion of pipe with depression rate of 1% (L = 20 mm).

**Figure 7 materials-15-06387-f007:**
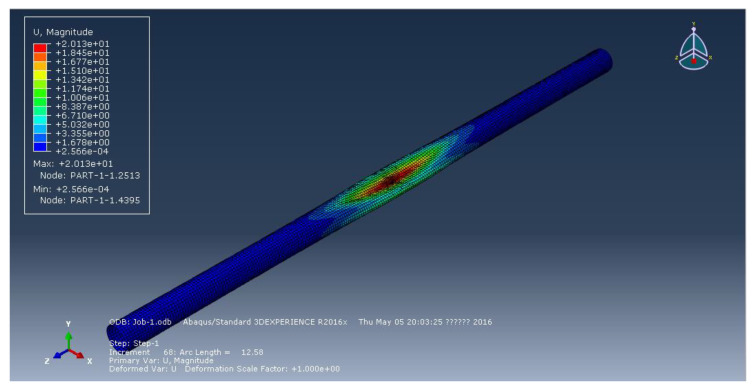
Distortion of pipe with depression rate of 1.5% (L = 20 mm).

**Figure 8 materials-15-06387-f008:**
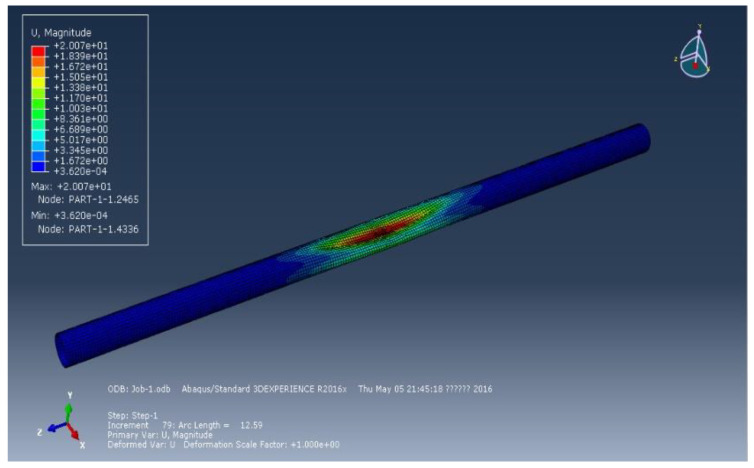
Distortion of pipe with depression rate of 2% (L = 20 mm).

**Figure 9 materials-15-06387-f009:**
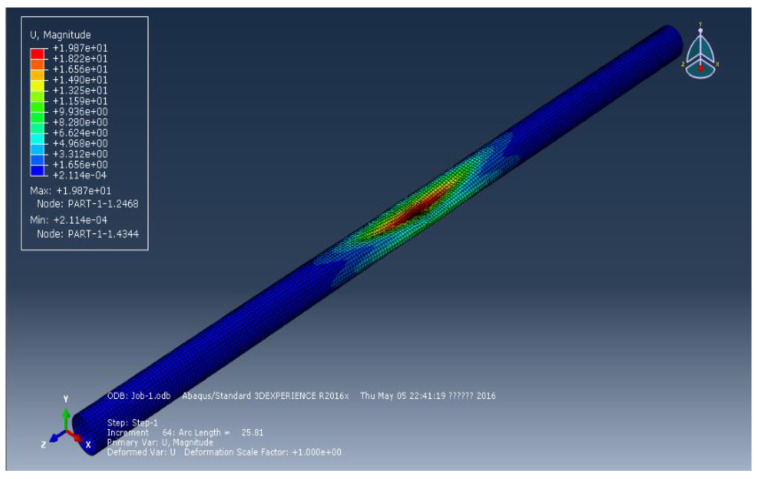
Distortion of pipe with depression rate of 3% (L = 20 mm).

**Figure 10 materials-15-06387-f010:**
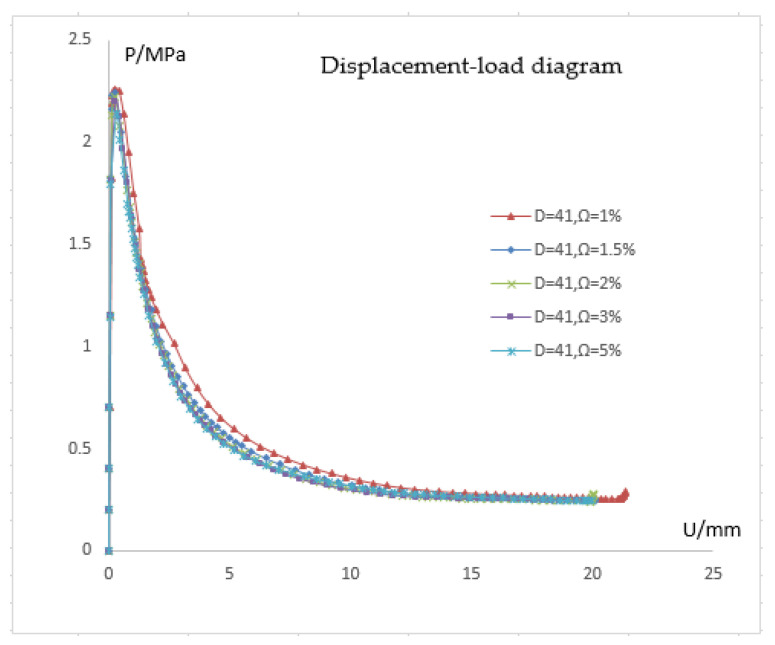
Pressure response of pipe with a depression length of 20 mm.

**Figure 11 materials-15-06387-f011:**
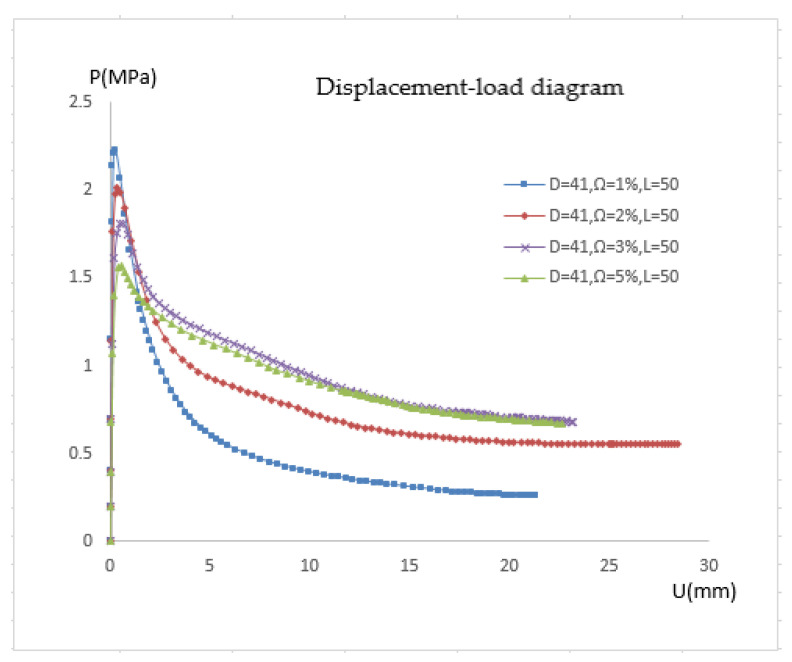
Pressure response of pipe with a depression length of 50 mm.

**Figure 12 materials-15-06387-f012:**
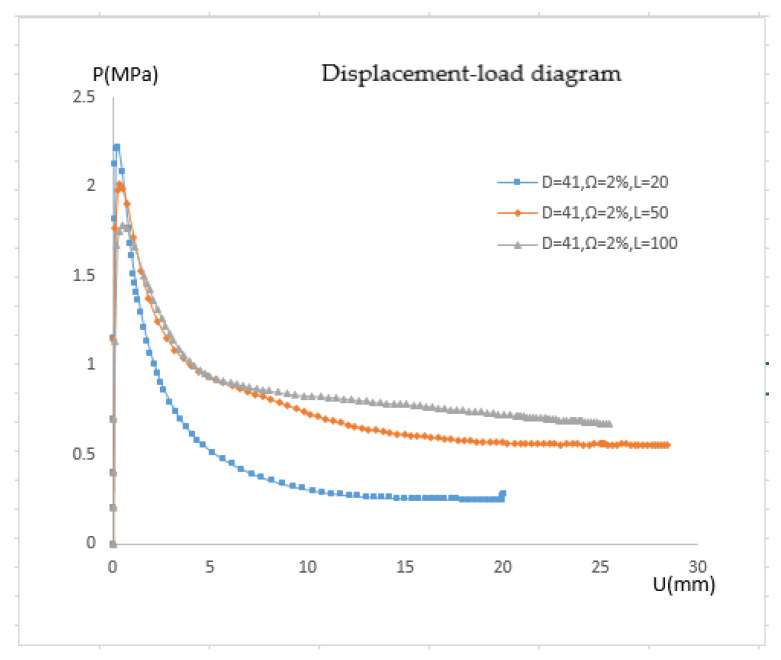
Pressure response of different pipelines with depression rate of 2%.

**Figure 13 materials-15-06387-f013:**
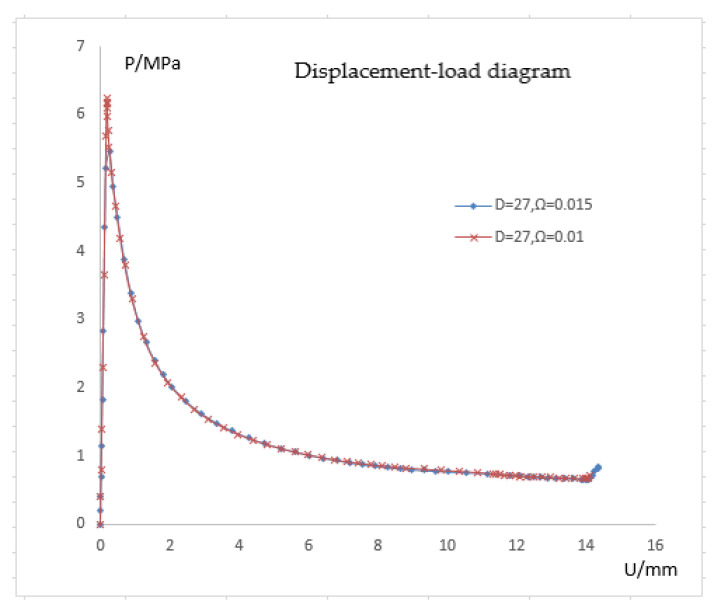
Pressure response of 27 mm radius-thickness ratio pipeline.

**Figure 14 materials-15-06387-f014:**
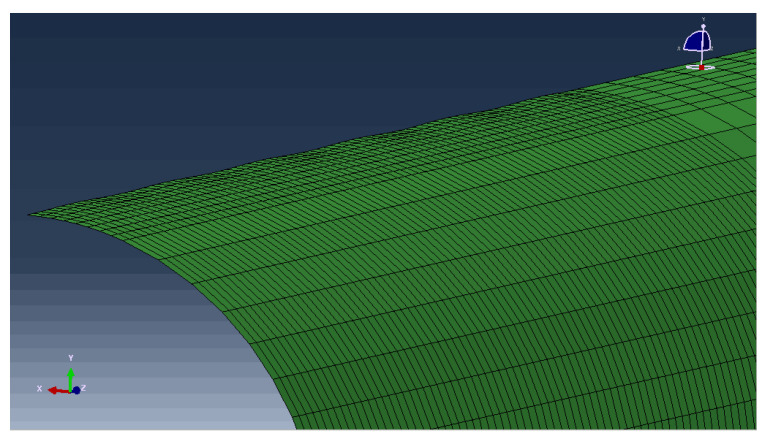
Local wrinkle of pipeline.

**Figure 15 materials-15-06387-f015:**
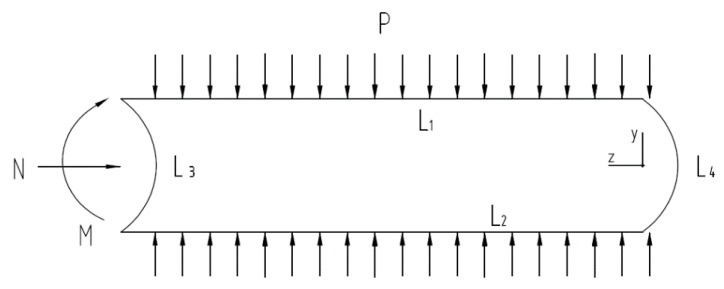
Schematic diagram of pipeline stress.

**Figure 16 materials-15-06387-f016:**
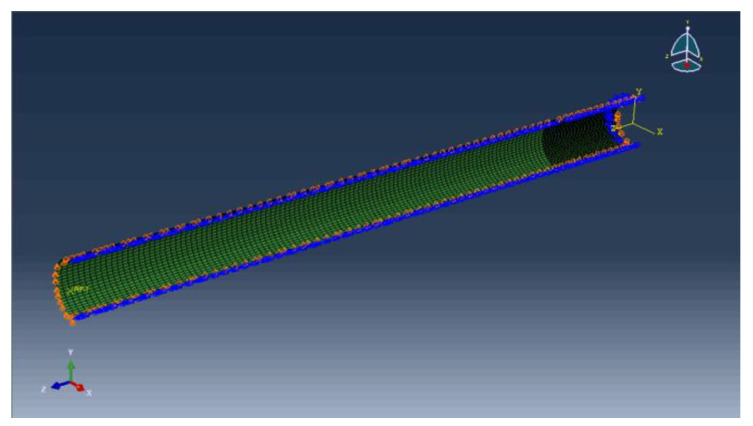
Boundary conditions for pipeline.

**Figure 17 materials-15-06387-f017:**
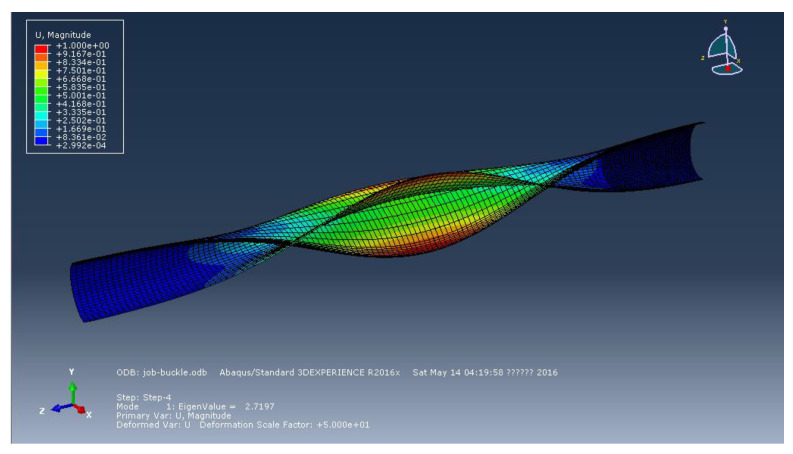
The first mode of pipeline.

**Figure 18 materials-15-06387-f018:**
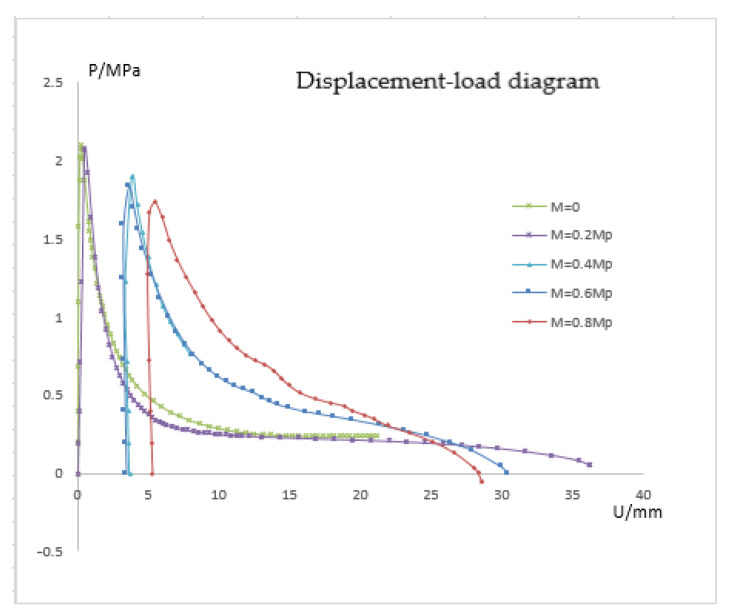
Pressure-displacement curves of pipelines under different bending moments.

**Table 1 materials-15-06387-t001:** Pipe parameter table.

$D_{0}$ $\left(mm\right)$	t $\left(mm\right)$	$\frac{D_{0}}{t}$	E $\left(\mathrm{GPa}\right)$	$E_{t}$ $\left(MPa\right)$	$\sigma_{y}$ $\left(MPa\right)$	$\frac{\sigma_{y}}{E}$	$P_{y}$ $\left(KN\right)$	$M_{p}$ $\left(KN\cdot mm\right)$
42	1	42	69	1500	90	0.0013	11.59	151.29
28	1	28	69	1500	90	0.0013	7.91	70.56

**Table 2 materials-15-06387-t002:** Critical pressure reduction caused by buckling interaction.

	M = 0	M = 0.2 Mp	M = 0.4 Mp	M = 0.6 Mp	M = 0.8 Mp
critical buckling pressure $P_{I}$	2.12 MPa	2.07 MPa	1.90 MPa	1.83 MPa	1.73 MPa
reduction rate	0	2.36%	10.38%	13.68%	18.40%

## Data Availability

Not applicable.

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
