# Peer review of "Numerical Simulation of Local Buckling of Submarine Pipelines under Combined Loading Conditions"

_materials, 2022, doi:10.3390/ma15186387_

Round 1
Reviewer 1 Report
The paper is well-written, but requires some comments to be addressed for the paper to be recommended for publication.
- Please expand the literature in your introduction to include significant literature positions in the area described in this paper (DOI):
10.1002/nme.6757 10.1016/j.compstruct.2015.06.058,
10.1016/j.compstruct.2015.11.018. - Please demonstrate the novelty of this paper at the end of the introduction.
- Please improve quality of the figures 2-6 and 10 for better readability.
- Please make a better presentation of the boundary conditions in Figure 9.
- There are very limited visualizations of the numerical simulation results in the paper. Please show more impressive results of numerical simulations.
- The conclusions are laconic. Please take care to provide a thorough description of the conclusions of the research results in the context of both quantitative and qualitative evaluation.
Reviewer 2 Report
Aug 17, 2022
Ref.: materials-1854813
Title: Numerical simulation of local buckling of submarine pipelines under composite load conditions
This article contains good and valuable information about the use of numerical modeling to examine the effects of initial geometric flaws and the diameter-to-thickness ratio on the local buckling of undersea pipelines.
In my opinion, several aspects should be modified or detailed more in-depth prior to publication, thus major modifications are advised. Here is a list of main comments: After correcting the manuscript based on the comments, I will announce my opinion regarding the acceptance or rejection of the article.
Consider all the comments below and highlight the changes.
1-In the abstract, it should have one sentence per each: context and background, motivation, hypothesis, methods, results, and conclusions. In the abstract, please add an indication of the achievements from your study that are relevant to the journal scope. Please be concise - maximum 1-2 lines. The abstract should state briefly the purpose of the research, the principal results, and major conclusions and also indicate the finding by the number or percentage of the calculations and improvements.
2-The introduction section, should follow the state of the art in this field and review what has been done, for supporting the research gap and the significance of this study. Please improve the state-of-the-art overview, to clearly show the progress beyond the state of the art. The lack of proper justification creates the wrong impression that the authors are unaware of the recent developments. In the introduction section, the author provides many citations (almost 30) and related works. However, what is the research gap? Which is not presented well. The concluding part of the introduction is not convincing enough. More critical discussion for better highlighting the novelty and significant observations from the study raised by the reviewer.
3-What was the main reason for choosing Unit selection? (Choosing the S4R?)
4- In section: 3.4 Boundary conditions, the authors mentioned that;
Because the pipeline's modeling length is 1000 mm, tens of times the diameter, and the constraint conditions at either end have little impact on the middle portion, one end restricts the circumferential displacement while the other restricts the axial displacement in order to facilitate. What is the reason of the authors for this claim?
5- In Figure 6. They claim that the critical pressure is around 19 percent lower than the theoretical value at 6.24 MPa when the depression rate is 0.01, and at 5.45 MPa when the depression ratio is 0.015. but it is not clear from the graph and both graphs look similar.
6- Please provide more related tables and compare your work with others in one table.
7- The conclusion is pretty generic and fails to provide any improvement in the existing knowledge base. Limitations in the suggested approach should be discussed in the conclusions section. In the other words, Conclusions must go deeper. Conclusions are not just about summarizing the key results of the study it should highlight the insights and the applicability of your findings/results for further work. Please make it more concise and show only the high-impact outcomes. All conclusions must be convincing statements on what was found to be novel, and impactful based on the strong support of the data.
8- Check the language of the article. In some parts, it needs to be corrected.
9-most of the references are not up to date (more than 70%) and mostly ranged from 1961 till 2010, please substitute/add up-to-date references.
Round 2
Reviewer 2 Report
Congratulations to all the authors
Author Response
We appreciate for editor and reviewer's warm work earnestly. In all, we found your comments are quite helpful. They help us for the further improvement. Thank you again and may the joy and happiness always be with you.